# Epigenetic Regulation of Myogenesis: Focus on the Histone Variants

**DOI:** 10.3390/ijms222312727

**Published:** 2021-11-25

**Authors:** Joana Esteves de Lima, Frédéric Relaix

**Affiliations:** Université Paris Est Creteil, INSERM, EnvA, EFS, AP-HP, IMRB, F-94010 Creteil, France; joana.esteves-de-lima@inserm.fr

**Keywords:** myogenesis, histone variants, PAX7, MYOD, H3.3, HIRA, H2A.Z, macroH2A

## Abstract

Skeletal muscle development and regeneration rely on the successive activation of specific transcription factors that engage cellular fate, promote commitment, and drive differentiation. Emerging evidence demonstrates that epigenetic regulation of gene expression is crucial for the maintenance of the cell differentiation status upon division and, therefore, to preserve a specific cellular identity. This depends in part on the regulation of chromatin structure and its level of condensation. Chromatin architecture undergoes remodeling through changes in nucleosome composition, such as alterations in histone post-translational modifications or exchange in the type of histone variants. The mechanisms that link histone post-translational modifications and transcriptional regulation have been extensively evaluated in the context of cell fate and differentiation, whereas histone variants have attracted less attention in the field. In this review, we discuss the studies that have provided insights into the role of histone variants in the regulation of myogenic gene expression, myoblast differentiation, and maintenance of muscle cell identity.

## 1. Introduction

### 1.1. Establishment of the Skeletal Muscle Lineage

Skeletal muscle tissue has a high regenerative potential that relies on tissue-specific stem cells, the satellite cells. These cells are characterized by the expression of the paired-homeobox transcription factor PAX7 and, in a subset of muscles, by the co-expression of its paralog PAX3 [1]. Muscle stem cells are established during fetal development where they adopt a satellite position under the basal lamina of the fiber and progressively initiate cell cycle exit into quiescence [2]. A pool of PAX3- and PAX7-double positive muscle stem cells constitutes a reservoir that allows fetal and postnatal muscle growth [3,4,5]. In the trunk and limb muscles, PAX3/7 lie genetically upstream of the myogenic regulatory factors (MRFs) that trigger cell commitment towards the myogenic lineage. The early-expressed MRFs include MYF5, MRF4 (MYF6), and MYOD, while Myogenin (MYOG) is expressed later during myoblast differentiation and is associated with terminal cell cycle exit [2]. The signaling cascades and the gene regulatory networks that operate to establish the myogenic lineage have been studied thoroughly but the epigenetic regulation of such processes is less understood.

### 1.2. The Epigenetic Regulation of Gene Expression

Skeletal muscle development and regeneration depend on the fine regulation of transcription factor activity and subsequent gene expression. The epigenetic status of a specific gene locus constitutes additional layers of regulatory events that control cell differentiation, fate, and identity. The epigenetic regulation of gene expression depends on the chromatin architecture, the histone post-translational modifications (PTMs), the nucleosome composition, and DNA methylation. In eukaryotic cells, nucleosomes are composed of 147 bp of DNA that wraps around the histone octamer complex, which contains a central tetramer comprising the histones H3-H4 and two external dimers of histones H2A-H2B [6,7]. The nucleosome is further stabilized by the presence of the linker histone H1. N-terminal tails of the histones and the C-terminal tail of the histone H2A extend beyond the nucleosome complex and are subjected to PTMs [8]. Histone tails can undergo PTMs such as acetylation, methylation, phosphorylation, and ubiquitylation, among others. Nucleosome remodeling is linked to several cellular processes such as DNA replication, DNA damage repair, transcriptional regulation, and maintenance of constitutive heterochromatin (centromere and telomeres) [6]. Based on these nuclear phenomena, the incorporation of histones into nucleosomes is characterized as being cell cycle dependent or independent, with distinct histone variants being deposited at each process. In mice, H2A, H2B, and H3 present several variants, while H4 has only one [6].

The remodeling of the nucleosomes is a tightly regulated process that can be associated with alterations in the type of histone variants and the ratio of old versus newly synthesized histones present in the nucleosome. Moreover, nucleosomes can be homotypic when comprising only one histone variant or heterotypic when more than one variant of the same histone is present [7]. Such characteristics can influence nucleosome stability and mobility. This reflects on the chromatin compaction levels that renders it more or less accessible to transcription factors, RNA polymerase II, and other factors required for transcription initiation. In addition, changes in histone variant composition can be accompanied by modifications in the balance of the histone-associated PTMs [7,8]. Mechanisms regulating histone PTMs have been shown to be crucial in the context of muscle differentiation. In skeletal muscle, components of the PRC2 Polycomb complex, which mediates the methylation of the histone H3 Lysine 27 (H3K27me3), such as EZH2 and SUZ12, are required for silencing myogenic differentiation genes in proliferating myoblasts [9,10]. *Ezh2* deletion specifically in satellite cells, further revealed its role in controlling stem cell self-renewal and maintaining myogenic gene expression and identity [11]. However, during muscle regeneration and in response to muscle inflammation signals, the PRC2 components EZH2 and YY1 allow satellite cell differentiation by being recruited to the *Pax7* locus and repressing its expression [12]. In contrast, the methyltransferase SUV39H1 that methylates histone H3 Lysine 9 (H3K9me3) and represses transcription, sustains the undifferentiated state of myoblasts by inhibiting late myogenic marker expression such as *Myog* and *Mck* [13,14]. The PRC1 Polycomb complex, which includes the ring finger protein BMI1, is recruited to chromatin regions marked by H3K27me3 and reinforces silencing through ubiquitylation of histone H2A [15]. Conditional activation of *Bmi1* expression in satellite cells of dystrophic mouse muscles (*mdx* mouse model that lacks Dystrophin) improves muscle strength [16]. This process is mediated by metallothionein 1 (MT1)-dependent resistance to oxidative stress in satellite cells in a dystrophic context, since similar activation of *Bmi1* in healthy muscles after traumatic injury does not ameliorate muscle strength [16]. The maintenance of the satellite cells quiescent state also relies on tight regulation of histone PTMs. The histone H4 Lysine 20 methyltransferase SUV4-20H1 promotes facultative heterochromatin formation (H4K20me2/3), i.e., chromatin that transits between a more or less compact state to regulate gene expression, in quiescent satellite cells to inhibit premature decompaction of the chromatin and expression of activation-related genes such as *Myod1* [17].

The function of histone modifier complexes and subsequent variations in histone PTMs have been widely studied in the context of transcriptional regulation and cell differentiation, in particular in myogenesis. In this review, we aim to focus exclusively on the studies that link histone variants and transcriptional regulation in myogenesis (Table 1). For detailed reviews on histone variants, histone chaperones, and histone functions besides myogenesis, we suggest the following comprehensive reviews to the readers [7,18].

## 2. The Role of the Non-Canonical h3 Histone Variant H3.3 in Myogenesis

### 2.1. The Histone H3 Family

The histone H3 family comprises two canonical, replication-dependent histone variants, H3.1 and H3.2, the centromeric-specific histone H3 variant CENPA, and the non-canonical histone variant H3.3. The replication-independent histone H3.3 differs from H3.1 and H3.2 by 5 and 4 amino acids (aa), respectively [7,19]. H3.3 incorporation into chromatin was identified as a requirement for the epigenetic memory of an active gene by stabilizing gene expression in somatic cells upon division [20]. In gene bodies H3.3 correlates with the H3K36me3 histone mark and is broadly associated with active transcription. Moreover, H3.3 enrichment in promoters and regulatory regions was mainly associated with active transcription in different cell types, given its preferential association with active histone marks like H3K4me3 and H3K9ac [21]. However, recent studies found that the recruitment of H3.3 into chromatin is associated with transcription regulation, either to activate or repress gene expression, when enriched in enhancers or promoters [22]. In mouse embryonic stem cells (mESC) the H3.3 histone variant is specifically deposited into the nucleosomes by the histone chaperone HIRA (gene bodies and transcription start sites (TSS)) and by DAXX in constitutive heterochromatin (telomeres and pericentromeric regions) [6,19]. Interestingly, recruitment of H3.3 to transcription factor binding sites (TFBS) is mostly independent of HIRA, which suggests involvement of other chaperones in the regulation of transcription [23,24].

### 2.2. H3.3 Function during Myogenesis

In skeletal muscle cells H3.3 plays a major role in cell differentiation. In primary cultures of chick myoblasts, the synthesis of the canonical histone variant H3.2 is predominant to that of H3.3 in proliferating culture conditions, consistent with H3.2 being replication-dependent. When triggered to differentiate, myoblasts stop replicating, fuse into myotubes, and H3.2 synthesis becomes undetectable while H3.3 continues to be synthesized, suggesting that H3.3 is required during myogenic differentiation [25]. At the RNA level, canonical H3.1 and H3.2 variant-encoding genes are expressed in myoblasts in growth culture conditions but not in differentiation, while H3.3-encoding genes are expressed in both growth and differentiating conditions [26]. In the mouse myoblast cell line C2C12, H3.3 incorporation in the promoter and the core enhancer region (CER) of *Myod1* is required for *Myod1* expression in differentiating myoblasts [26]. In addition, the histone chaperone HIRA and its cofactor ASF1A mediate H3.3 recruitment to the *Myod1* locus. The loss of H3.3, HIRA, or ASF1A leads to decreased *Myod1* expression levels and blocks the differentiation potential of the myoblasts that fail to fully differentiate and form myotubes (Figure 1A). Mechanistically, the enrichment of H3.3 at *Myod1* regulatory regions promotes the transition towards a more permissive chromatin state that facilitates the recruitment of the RNA polymerase II [26]. The HIRA-ASF1A complex further regulates myogenic differentiation by interacting with the muscle-specific transcription factor MEF2, which is required for the activation of MEF2 target gene expression [27]. However, putative changes in histone variant incorporation into chromatin were not addressed in this context. The phosphorylation of HIRA by the AKT1 kinase was proposed to mediate HIRA function as an H3.3 chaperone [28]. The levels of phosphorylated HIRA are high in proliferating C2C12 myoblasts, which limits the expression of myogenic genes, while dephosphorylation is required for H3.3 deposition and gene expression activation upon differentiation [28]. Interestingly, MYOD regulates myogenic gene expression by interacting with the ubiquitous chromodomain helicase DNA-binding domain 2 (CHD2) protein to deposit H3.3 in myogenic gene loci prior to differentiation in C2C12 [29]. The knockdown of *Chd2* or *Myod1* prevents H3.3 incorporation in myogenic genes but not in housekeeping genes (Figure 1B) [29]. This suggests that tissue-specific transcription factors play a role in the epigenetic regulation of target gene transcription by recruiting ubiquitous chromatin regulators. CHD2 is also part of the H3.3 incorporation complex in developmentally regulated gene loci of mESC, being essential to prevent suppressive chromatin formation at these genomic regions. In *Chd2*-depleted mESC, there is a decreased H3.3 deposition and increased enrichment of the histone repressive mark H3K27me3 at developmental gene loci [30]. Consequently, there is a blockage in mESC differentiation into the different germ layers. These studies indicate that CHD2 and H3.3 are associated with tissue differentiation in both pluripotent and somatic cells.

While H3.3 is required for myogenic gene expression and myoblast differentiation (Table 1), canonical H3 variants are associated with proliferating myoblasts [25,26]. Moreover, HIRA, ASF1A, and H3.3 expression levels decrease to half when C2C12 are forced to differentiate into osteoblasts but are maintained during myoblast differentiation, which correlates H3.3 with muscle cell differentiation [31]. The forced expression of H3.1-GFP fusion protein in C2C12 myoblasts in differentiation culture conditions leads to an impaired expression of myogenic genes, a decreased number of MYOG-positive cells, and less myotube formation [32]. H3.3 chromatin immunoprecipitation followed by sequencing (ChIP-seq) showed that in the context of H3.1-GFP overexpression there is a decrease in H3.3 incorporation preferentially at skeletal muscle genes promoters. The changes in nucleosome composition regarding the type of histone H3 variants further disturbs the balance of the histone PTMs. This is also the case in skeletal muscle gene loci, where H3.3-GFP nucleosomes are enriched with the active transcription-associated histone mark H3K4me3, while H3.1-GFP nucleosomes are preferentially linked to the repressive transcription mark H3K27me3 [32]. The observed differences in PTMs is associated with the recruitment of the histone methyltransferase EZH2 to myogenic gene loci in the presence of the canonical variant H3.1 [32]. This highlights the important role of histone variant regulation in nucleosome composition to maintain the balance of PTMs and control gene expression.

### 2.3. H3.3 Genetic Diversity

H3.3 is encoded by two genes, *H3f3a* and *H3f3b,* which, in contrast to canonical histone-encoding genes, contain introns and a poly-A tail signal [33]. Remarkably, these two genes encode for an identical protein sequence. H3.3 is essential for mouse embryogenesis since double knockout embryos for *H3f3a* and *H3f3b* are lethal at early developmental stages (between E3.5 and E6.5) [34]. At the cellular level, loss of H3.3 suppresses cell cycle and leads to cell death. Single knockout mice for *H3f3a* are viable with males being subfertile, while *H3f3b* knockouts are growth-deficient, dying at birth [35]. In contrast, another report described the single *H3f3b* knockout mice to be normal and fertile [34]. This shows the existence of developmental compensation between the two genes with *H3f3b* possibly having a more functional role given the reported postnatal lethality phenotype [35]. While H3.3 was traditionally thought to be the only non-canonical H3 histone variant, a recent study identified 14 previously uncharacterized H3 genes in the mouse genome, 13 of which are related to H3.3 and are capable to stably integrate into nucleosomes when their expression is forced in C2C12 cells [36]. C2C12 myoblasts endogenously express the H3 subvariants H3mm7, H3mm8, H3mm13, and H3mm15. In particular, H3mm7 overexpression positively enhances skeletal muscle gene expression and differentiation in C2C12 cells [36]. Conversely, deleting *H3mm7* by Crispr/Cas9 technology in C2C12 cells leads to the opposite phenotype, i.e., an inhibition of myoblast differentiation [37]. *H3mm7*-null mice are viable and indistinct from the wild type counterparts but muscle regeneration is delayed in these mice [37]. In adult mouse skeletal muscles, *H3mm7* is expressed in satellite cells, with a stronger expression associated with the quiescent state, but not in myofibers. The delay in regeneration of *H3mm7*-null mouse muscles is associated with an increased number of embryonic myosin heavy chain (MYH3)-positive fibers, a higher number of fibers with smaller cross-sectional area (CSA), and the downregulation of fiber maturation-related genes [37]. The number of satellite cells was not affected, which suggests that H3mm7 is dispensable for the maintenance of the stem cell pool but required for myoblast differentiation and maturation. Analysis in C2C12 cells showed that mechanistically, when enriched in myogenic regulatory regions, H3mm7 promotes gene expression by enhancing chromatin accessibility, as observed by assay for transposable accessible chromatin followed by sequencing (ATAC-seq) [37]. In fact, nucleosomes containing the subvariant H3mm7, that differs from H3.3 by only 2 aa, are more unstable as observed by crystal structure analysis and by NaCl dissociation. The increased instability of H3mm7-containing nucleosomes makes them more mobile, i.e., leads to increased rate of histone exchange, observed by fluorescence recovery after photobleaching (FRAP), when compared to H3.3-containing nucleosomes [37].

### 2.4. Role of the Histone Chaperone HIRA in Myogenic Cells

Since the histone chaperone HIRA specifically incorporates H3.3, some studies focused on the role of HIRA in myogenesis. *Hira*-null mouse embryos show an early defective development of the mesodermal tissues, associated with an arrested gastrulation and lethality at E9.5, making it impossible to further analyze the skeletal muscle phenotype [38]. The use of a conditional mouse line with the *Hira* floxed allele combined with the *Mrf4^Cre^* allele allowed a myoblast-specific deletion of HIRA [39]. Animals lacking HIRA in myoblasts do not present any defects at 6 weeks of age but at 6 months, the myofibers become hypertrophic with sarcolemmal perforation and oxidative damage. This is associated with increased expression of myogenic genes and decreased expression of genes linked to cellular stress response pathways [39]. In addition, the number of oxidative type I fibers (MYH7-positive) and the expression of *Myh7* gene is increased in the *tibialis anterior* (TA) muscle at 6 months of age. Increased *Myh7* gene expression was also observed when HIRA was deleted in cardiomyocytes [40].

H3.3 is incorporated in the regulatory regions and promoters of myogenic genes prior to differentiation and its recruitment increases during differentiation (Table 1) [26,28,29]. In addition, H3.3 was linked to myogenic and neurogenic cell differentiation [26,29,32,41], but the functional role of H3.3 in progenitor cells prior to differentiation has received less attention. Recently, a study using a conditional and inducible knockout of *Hira* in muscle stem cells (PAX7-positive cells) has identified H3.3 as being essential to maintain the myogenic identity of stem cells [42]. Satellite cells lacking HIRA and H3.3 enrichment lose muscle stem cell- and myogenic-related gene expression, while activating the expression of genes from alternative lineages. At a functional level, these muscles are unable to regenerate; there is an increased number of immature MYH3-positive fibers associated with a decreased CSA and loss of satellite cells [42]. Mechanistically, in the absence of HIRA, H3.3 enrichment of myogenesis-related genes is lost, which is associated with a reduction in H3K27ac active histone mark and a decrease in chromatin accessibility, resulting in reduced expression levels of genes essential for maintaining myogenic identity (Figure 1C) [42]. This shows that H3.3 enrichment at myogenic loci is essential for muscle identity and maintenance of the satellite cell pool. Moreover, the use of self-labelling protein tag (SNAP-Tag) H3.3-reporter mice showed that fate decision of muscle stem cells is linked to H3.3 symmetric distribution in the daughter cells [43].

### 2.5. Primate-Specific H3 Variants H3.X and H3.Y

The H3 variants H3.X and H3.Y [44] are expressed in several human cell lines and are incorporated into chromatin in a similar fashion as H3.3; however, H3.Y-containing nucleosomes are associated with more relaxed chromatin and active transcription [44,45]. Moreover, H3.Y interacts with and is deposited by the HIRA complex, but not the DAXX complex, which is consistent with its incorporation in the TSS of actively expressed genes rather than heterochromatin [45,46]. DUX4 is a double homeobox transcription factor that when misexpressed in skeletal muscle promotes a toxic cellular environment and causes facioscapulohumeral muscular dystrophy (FSHD) [47]. Myoblasts derived from FSHD muscle biopsies showed that *DUX4*, *H3.X,* and *H3.Y* are upregulated compared to control samples. Consistently, a human myoblast cell line with an induced DUX4 transgene displays increased *H3.X* and *H3.Y* expression levels and incorporation of these histone proteins in the gene bodies of DUX4 target genes [48]. Moreover, silencing *H3.X* and *H3.Y* expression with small interfering RNAs (siRNAs) upon DUX4 induction in myoblasts prevents the upregulation of DUX4 target genes, such as *ZSCAN4* and *TRIM43* [48]. This suggests that H3.X and H3.Y incorporation at DUX4 target gene loci contributes to the persistence of deleterious expression of those genes, leading to a FSHD transcriptional phenotype.

## 3. H2A Histone Variants and Myogenic Gene Expression

### 3.1. The H2A Family

The histone variants from the H2A family display higher variation among their sequences compared to the H3 family. The H2A family is composed of several replication-independent variants besides the canonical H2A, such as H2A.Z, H2A.X, H2A.Bbd, and macroH2A (mH2A) [7,49].

H2A.Z histone variant is encoded by two genes, *H2afz* and *H2afv,* that originate two distinct protein isoforms differing by 3 aa, H2A.Z-1, and H2A.Z-2, respectively [50]. The H2A.Z histone variant is mostly enriched at TSS, promoters, enhancers, facultative heterochromatin, and centromeres, and was linked to both transcriptional activation and repression, in which H2A.Z-associated PTMs play a major role [51]. H2A.Z is expressed in C2C12 myoblasts and silencing its expression does not interfere with myoblast differentiation or myotube formation, which shows that its continued expression is not required for differentiation [52]. A non-acetylatable form of H2A.Z (where the 5 tail lysines are mutated into arginines) fused to GFP (H2A.Z-Ac-mut-GFP) can be incorporated into the genome in a similar fashion as the wild type H2A.Z fused to GFP (H2A.Z-GFP), including at the *Myod1* and *Myog loci* (Figure 2A). H2A.Z-Ac-mut-GFP-expressing myoblasts display reduced myogenic gene expression, such as *Myod1*, *Myog,* and *Myh3*, when triggered to differentiate. Impaired myogenesis is associated with the lack of RNA polymerase II recruitment in the presence of H2A.Z-Ac-mut-GFP to myogenic gene loci upon differentiation [52]. Consequently, H2A.Z-Ac-mut-GFP overexpression in C2C12 cells blocks myotube formation while H2A.Z-GFP overexpression does not significantly interfere with this process, which shows the H2A acetylation regulates differentiation [52]. This shows that acetylation of the histone variant H2A.Z plays a role in transcription initiation of myogenic gene expression. The exchange of H2A for its variant H2A.Z is modulated by the chromatin remodeling complex SNF2-related CBP activator protein (SRCAP), a mechanism conserved in yeast and in mammals [53,54,55]. ChIP-seq analysis of C2C12 cells confirmed the presence of p18^Hamlet^ (ZNHI1), a component of the SRCAP complex and a substrate of the p38 MAPK pathway, at the *Myog* promoter in differentiating C2C12 cells [53]. In addition, p18^Hamlet^ is required for the incorporation of H2A.Z at the *Myog* promoter, and the enrichment of these two proteins at this genomic region increases during differentiation of C2C12 cells and murine primary myoblasts. Phosphorylation of p18^Hamlet^ by p38 is required for its recruitment, and for the incorporation of H2A.Z in the *Myog* promoter, which suggests that p38 MAPK-dependent signals can impact on chromatin structural changes [53]. Inhibition of the expression of components of the SRCAP complex leads to impaired myogenic gene expression and blocks myoblast differentiation [53]. Myofibroblast differentiation that relies on TGF-β1 expression is also regulated by H2A.Z occupancy [56]. In contrast, in this case, in order to facilitate TGF-β1 expression, H2A.Z must be depleted from the TGF-β1 promoter region through a mechanism that requires the ribosomal function regulator ELF6 (Yang et al., 2015).

H2A.Z is crucial for embryonic development and *H2A.Z*-null embryos die just after the blastocyst stage, between E4.5 and E6.5 [57]. In mouse TA muscles, ChIP-seq analysis revealed that H2A.Z is enriched in TSS and regulatory regions of actively transcribed myogenic genes [58]. However, specific ablation of H2A.Z-1 and H2A.Z-2-encoding genes in post-mitotic myoblasts, using the human α-skeletal actin promoter conditional (*HSA^Cre^*) allele has no effect on muscle homeostasis or regeneration [58]. Moreover, RNA-seq analysis from control and H2A.Z-depleted muscles show that in vivo inactivation of H2A.Z is not required to maintain or activate transcription in a post-mitotic cellular state (Table 1). While H2A.Z is required for gene expression regulation in mitotic cells, namely in mouse embryonic fibroblasts (MEFs), it is dispensable for transcription regulation of terminally differentiated myoblasts [58]. In vascular smooth muscle cells, H2A.Z occupies genomic regions near VSMC-related genes and is required to maintain their expression [59]. This occurs by H2A.Z-dependent recruitment of SMAD3, an important factor that promotes vascular smooth muscle cell differentiation, and the mediator of RNA polymerase II transcription MED1. The enrichment of H2A.Z at these genomic loci is reduced in VSMC undergoing dedifferentiation, which occurs after several cell passages and in diseased human vascular tissues [59]. H2A.Z recruitment to vascular smooth muscle cell-related gene loci, and its role in the positive regulation of the transcription of these genes, show that H2A.Z is required for the maintenance of smooth muscle cell identity and contributes to cell fate decisions [59].

### 3.2. The mH2A Family

The mH2A subfamily of histone variants is about three times the size of the canonical H2A due to the presence of an evolutionary conserved non-histone globular macrodomain, described to be associated with X chromosome inactivation, transcriptional repression, and reprogramming inhibition [60,61,62,63]. The transcriptional repression activity is associated with increased stability of heterotypic mH2A-H2B-containing nucleosomes compared with canonical H2A-H2B-containing nucleosomes [64]. Three distinct mH2A histones have been identified, mH2A1.1 and mH2A1.2, which are two isoforms generated from alternative splicing of the *H2afy* gene, and mH2A2 that is encoded by *H2afy2* [65,66,67]. RNA-seq analysis revealed that in C2C12 myoblasts the mH2A1.2 isoform is more expressed than mH2A1.1 in growth conditions, while in myotubes both isoforms are expressed at similar levels [68]. The comparable levels of expression for both isoforms in myotubes is the consequence of a switch in splicing that occurs after myoblast differentiation and leads to the decrease of mH2A1.2 transcript levels and the increase in mH2A1.1 [69]. Silencing of mH2A1.2 with siRNAs does not affect C2C12 myoblasts in growth conditions but inhibits *Myog* expression and myotube formation when triggered to differentiate [68]. Moreover, gene ontology (GO) analysis of RNA-seq data revealed terms associated with muscle cell development and differentiation to be downregulated in mH2A1.2 siRNA-transfected C2C12 cells. In the same silencing conditions, ChIP-seq analysis showed that there is a specific loss in the enrichment of the active transcription-associated histone mark H3K27ac in myogenic-specific promoters and enhancers, which shows the requirement of mH2A1.2 at these loci to maintain H3K27 acetylation and active gene expression (Figure 2B) [68]. The recruitment of the homeodomain-containing transcription factor PBX1, required for the MYOD-dependent activation of *Myog* expression, to muscle development-related gene loci is also regulated by mH2A1.2 prior to differentiation (Figure 2B) [68]. The *mH2A1* and *mH2A2* double knockout mice have impaired prenatal and postnatal development, which is associated with the mH2A function in regulating metabolic-related gene expression in the liver [70]. Moreover, mice lacking the histone variant mH2A1 are viable and fertile [71]. In both cases, analyses to address skeletal muscle defects in these mice have not been described. However, when myoblasts obtained from muscles of *mH2A1*-null mice are cultured *in vitro*, they differentiate and express MYH3 but lack the ability to form large myotubes [72]. Both mH2A1 isoforms regulate myoblast fusion, however with distinct outcomes (Figure 2C) [72]. Specific inhibition of each of the mH2A1 isoforms in C2C12 cells with siRNAs revealed that while mH2A1.1 promotes myoblast fusion, mH2A1.2 inhibits it [72]. This phenotype is linked to the opposite regulation of genes associated with GO terms such as extracellular matrix organization, cell adhesion, and skeletal system development. ChIP-seq analysis identified the mH2A1.1 isoform to be enriched at fusion-related genes when C2C12 are triggered to differentiate, which links mH2A1.1 to transcription activation and myoblast fusion [72].

### 3.3. H2A.X Histone Variant

The H2A.X histone variant is largely used as a marker of DNA damage when phosphorylated at serine 139 (S139) as a response to the induction of DNA double-strand breaks, but its role as transcriptional regulator remains poorly understood [7]. Consistently, H2A.X-null mice are growth retarded, radiation sensitive, and males are infertile [73]. The phosphorylated form of the H2A.X variant (γH2A.X) is used together with other indicators, such as p16 levels and β-Gal staining, to identify cellular senescence [74]. Cultured satellite cells isolated from old mouse muscles or old human biopsies have increased γH2A.X and p16 levels compared to young satellite cells [75,76]. Therefore, senescence has been linked to sarcopenia, which is age-related muscle loss. However, further analysis of human muscle biopsies showed that in skeletal muscles, old and young muscle cells, including satellite cells, have comparable γH2A.X numbers [77]. In contrast, γH2A.X-positive cells were increased in obese versus lean muscle biopsies, particularly in post-mitotic fiber nuclei. Moreover, RNA-seq analysis revealed upregulation of GO terms related to DNA damage and senescence to be linked to obese muscle biopsies [77]. These data illustrate that γH2A.X histone variant can be a good marker to detect senescence within muscle cell populations.

## 4. The Linker Histone H1 Variants and Myoblast Differentiation

In mammals, the linker histone H1 family comprises seven somatic variants (H1a to H1e, H1x and H10) and four germ cell-specific variants (testis: H1t, H1t2 and HILS1 and oocyte: H1oo) [78]. Single or double-deletions of H1 somatic variant genes do not impact mouse development but mutations in at least three genes lead to embryonic lethality, suggesting a threshold compensatory mechanism [79]. In particular, the triple-H1-null embryos for *H1c^−/−^;H1d^−/−^;H1e^−/−^* die at mid-gestation and mESC with the same combined mutations are more resistant to spontaneous differentiation, and the derived embryoid bodies fail to form the three germ layers [79,80].

Mass spectrometry analysis showed that the H1 histone variant H1b interacts directly with the myogenic transcriptional repressor msh homeobox 1 (MSX1) [81]. *Msx1* is expressed in proliferating myoblasts and is a potent inhibitor of myoblast differentiation in vivo and in vitro, by directly binding to the *Myod1* CER and repressing its transcription [81,82,83]. ChIP analysis revealed that MSX1 enrichment at the CER is associated with H1b recruitment to this genomic region. Moreover, silencing *H1b* expression in C2C12 cells with siRNA abrogates MSX1-dependent inhibition of myogenic differentiation (Figure 3A) [81]. In vivo analysis using *Xenopus* embryos showed that MSX1 overexpression inhibits myogenic gene transcription, and that the combined ectopic expression of MSX1 and Hb1 has a synergistic effect in the inhibition of these genes. This effect is specific to the MSX1-H1b interaction since overexpression of MSX1 with another H1 variant, such as H1e, does not lead to a combined inhibition of myogenic gene expression [81]. These results demonstrate that the structural nucleosome linker histone variants have additional roles in the regulation of transcription and cell differentiation. Inhibition of the scaffold protein SH2B adaptor protein 1 (SH2B1) expression by short hairpin RNA (shRNA) in C2C12 cells does not affect proliferation in growth culture conditions but reduces MYOG and MYHC protein levels and impairs myotube formation during differentiation [84]. Mass spectrometry and co-immunoprecipitation analysis revealed that SH2B1 interacts with several H1 histone variants and that this interaction decreases with myoblast differentiation [84]. During differentiation, H1 and SH2B1 enrichment is decreased at myogenic loci. However, histone H1 ChIP analysis showed that in C2C12 cells lacking SH2B1, H1 enrichment is retained in the promoter and regulatory regions of myogenic genes such as *Igf2* and *Myog* (Figure 3B) [84]. This is associated with the maintenance of the repressive histone mark H3K9me3 at the expense of the active mark H3K4me3 at muscle-related loci, which inhibits gene expression [84].

## 5. Conclusions

The role of the histone variants in myogenesis and more generally in the regulation of somatic cell identity and differentiation has relatively recently started to be unveiled. In contrast, studies on histone variants in pluripotent or embryonic stem cells have greatly contributed to describe the function of specific histone variants in maintaining pluripotency and/or regulating developmental gene expression upon differentiation. In addition, the use of pluripotent cell lines to study histone variants further contributed to uncovering the genomic enrichment of histone variants, the histone chaperones mediating histone variant deposition, or the mechanisms by which specific histone variants interfere with transcriptional modulators to regulate gene expression. With this review and the different studies that combine histone variants and myogenesis, we can highlight the importance of the H3 histone variant H3.3 and the H2A histone variants H2A.Z and mH2A1.2 in promoting myogenic gene expression and differentiation. These histone variants are enriched at myogenic gene loci during differentiation and they either recruit transcriptional regulators or undergo specific post-translational modifications required to maintain or activate gene expression. Studies on H1 variants, such as H1b, further show that replacement histones from distinct families can function as transcriptional regulators and control myogenic differentiation. This suggests that there is a broad regulation of gene expression by distinct histone variants; however, whether the role of each histone variant is specific to muscle cells or whether similar or contrasting functions operate in other somatic cells remains to be studied.

The field of somatic cell differentiation started recently to give more attention to the function of histone variants in transcriptional regulation, but has largely relied on in vitro studies. This is associated with the need to use fusion and/or tagged proteins, or the requirement of high cell numbers to accomplish specific experiments, where cell lines provide easier manipulation and virtually no limit of material. With the constant update of the technologies and protocols to perform genome-wide analysis with smaller inputs, it might be easier in the near future to combine epigenetic studies with in vivo approaches. Techniques such as single cell RNA-seq (scRNA-seq) combined with scATAC-seq start to make it possible to study gene expression and chromatin accessibility at the single cell level and can be performed on in vivo cells or tissues. Moreover, ChIP-seq improved techniques such as Cleavage Under Targets and Tagmentation (CUT&Tag) are now available for use with small numbers of cells and should lead to the identification of histone or histone mark enrichment from in vivo cells.

## Figures and Tables

**Figure 1 ijms-22-12727-f001:**
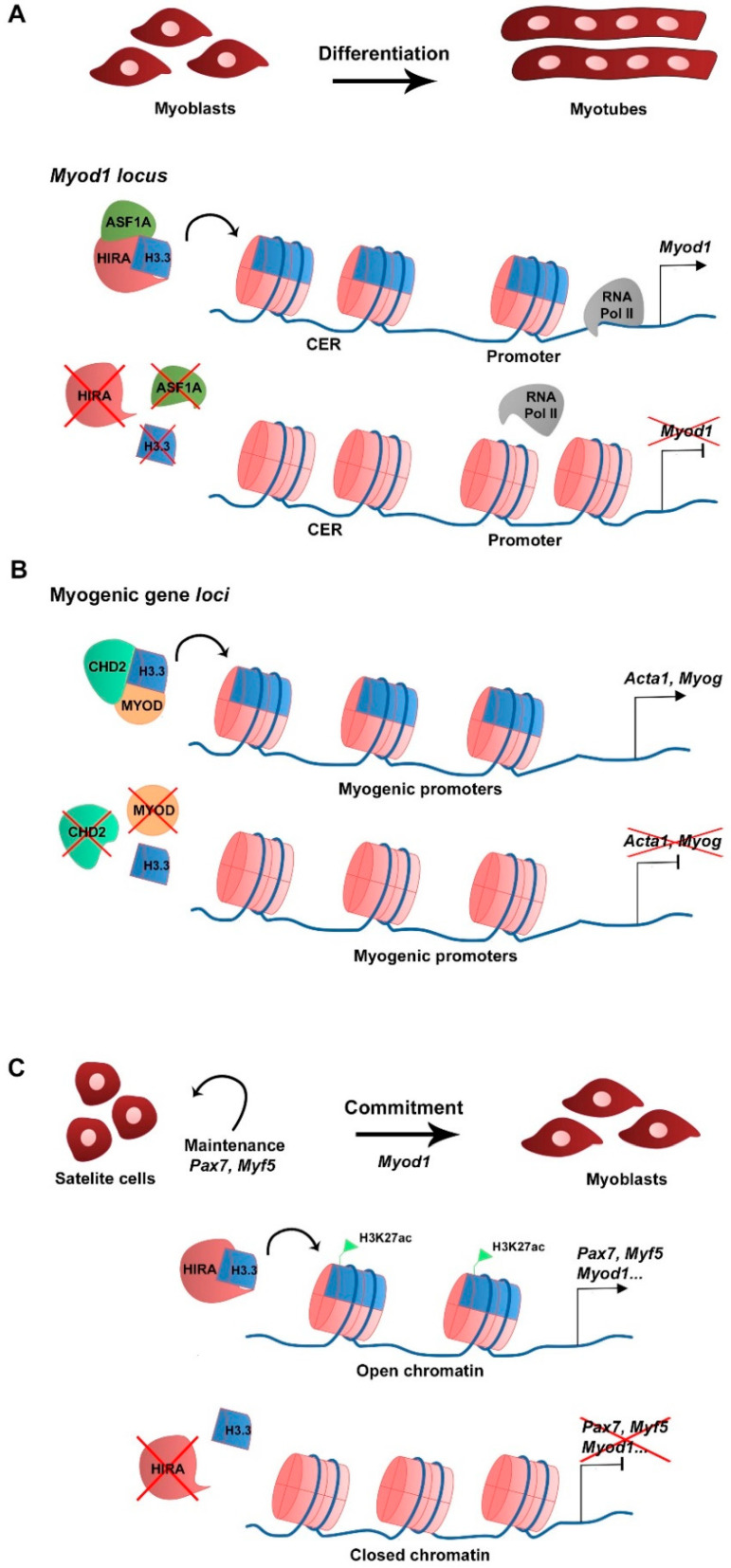
Schematic representation of H3.3 function in myogenesis. (**A**) At the myoblast stage *Myod1* expression is regulated by H3.3 deposition in the vicinity of the CER and promoter by the HIRA-ASF1A complex (Yang et al., 2011). (**B**) To allow differentiation and myotube formation, CHD2 directly interacts with MYOD to deposit H3.3 in promoters of myogenic differentiation-related genes and activate transcription (Harada et al., 2012). (**C**) Upon muscle injury, activated satellite cells require H3.3 deposition by HIRA in myogenic gene regulatory regions to maintain cell identity (*Pax7*, *Myf5* expression) and commitment (*Myod1*). H3.3 enrichment correlates with that of H3K27ac and with an open chromatin state, which favors myogenic gene expression (Esteves de Lima et al., 2021).

**Figure 2 ijms-22-12727-f002:**
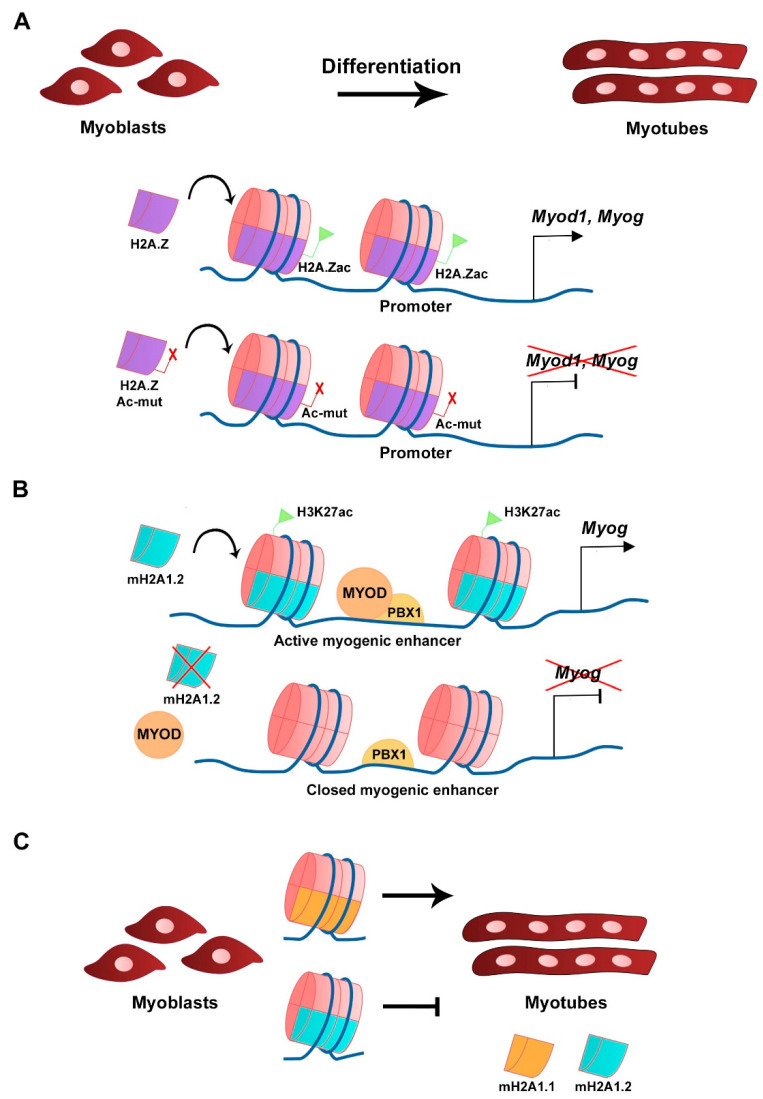
Histone H2A variants in myogenesis. (**A**) Acetylation of H2A.Z variant at the CER is required for *Myod1* expression and myoblast differentiation. Overexpression of a mutated and non-acetylatable form of H2A.Z inhibits *Myod1* expression (Law and Cheung, 2015). (**B**) The mH2A1.2 variant is required for myogenic enhancer activation prior to differentiation and correlates with H3K27ac histone mark. mH2A1.2 enrichment allows MYOD-PBX1 complex formation at the *Myog* promoter, activating transcription (Dell’Orso et al., 2016). (**C**) Distinct mH2A1 isoforms have different roles on myoblast differentiation (Hurtado-Bagès et al., 2020).

**Figure 3 ijms-22-12727-f003:**
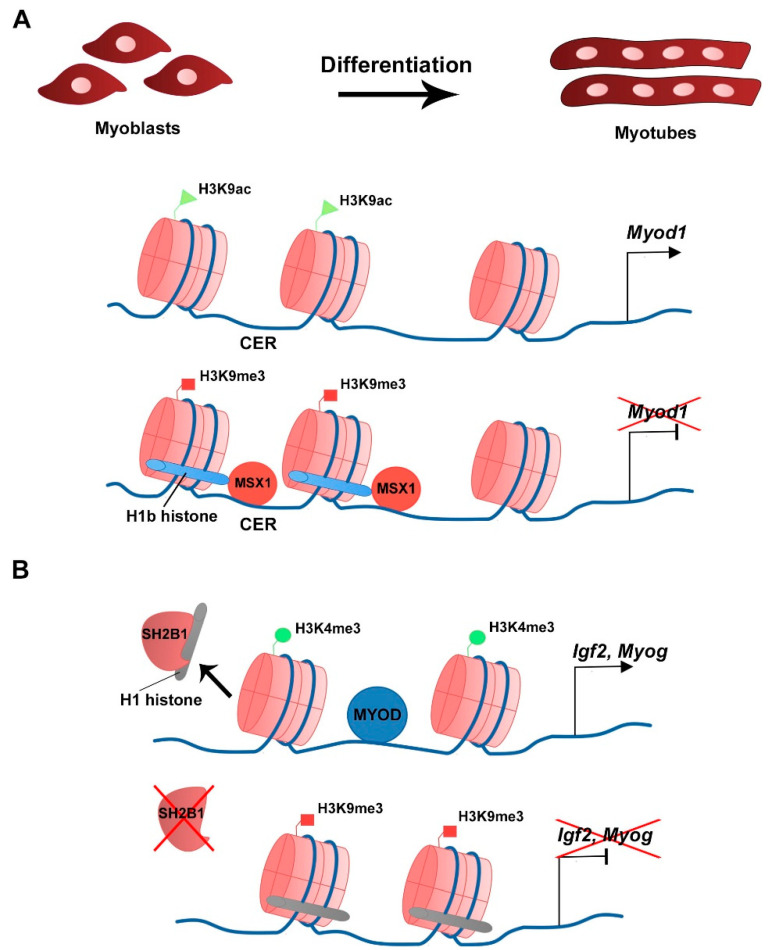
Linker histone H1 variants and myogenic gene expression. (**A**) At the myoblast growth stage Myod1 expression is inhibited by the presence of MSX1 that interacts with the H1 variant H1b at the CER, which correlates with the repressive histone mark H3K9me3 (Lee et al., 2004). (**B**) Upon differentiation, SH2B1 interacts and releases H1 histone variants from the regulatory regions of Igf2 and Myog. Lack of SH2B1 maintains H1 variants in the nucleosome that associate with the repressive mark H3K9me3 (Chen et al., 2017).

**Table 1 ijms-22-12727-t001:** List of the histone variants that regulate myogenesis.

Histone Variant	Expression/Deposition/Function	Reference
H3.3	Synthesis (protein) increases during chicken primary myoblast differentiation.	Wunsch and Lough, 1987
	Expression (mRNA) in C2C12 cells during growth and differentiation. Deposition by HIRA/ASF1A in Myod1 CER in C2C12 cells. Required for Myod1 expression and myoblast differentiation.	Yang et al., 2011
	Deposition by CHD2/MYOD complex in myogenic gene loci in C2C12 cells. Required for Myog expression and myoblast differentiation.	Harada et al., 2012
	Deposition in myogenic gene loci is associated with the histone mark H3K4me3.	Harada et al., 2015
	Hira KO in satellite cells leads to decreased H3.3 at myogenic gene loci. Association with the histone mark H3K27ac and accessible chromatin. Required for myogenic gene expression, regeneration and satellite cell identity.	Esteves de Lima et al., 2021
H3mm7	Expression (mRNA) in satellite cells in vivo and in C2C12 cells. Promotion of myogenic gene expression and differentiation. Required for normal regeneration.	Harada et al., 2018
H3.X, H3.Y	Deposition in regulatory regions of DUX4 target genes in FSHD. Activation of DUX4 target gene expression.	Resnick et al., 2019
H2A.Z	Deposited at myogenic genes promoters in primary myoblasts and C2C12 cells. Deposition is p38 MAPK-dependent and enriched during differentiation. Required for myogenic gene expression and myoblast differentiation.	Cuadrado et al., 2010
	Acetylation of H2A.Z is required for Myod1 expression and C2C12 myoblast differentiation. Acetylation of H2A.Z is required for RNA Pol II recruitment to myogenic gene loci.	Law and Cheung, 2015
	Enrichment in actively transcribed myogenic genes in vivo. Dispensable for gene expression, homeostasis and regeneration in post-mitotic fibers.	Belotti et al., 2020
mH2A1.1	Expression (mRNA) in C2C12 cells during growth and differentiation.	Dell’Orso et al., 2016 Posavec Marjanović et al., 2017
	Promotion of myoblast fusion in C2C12 cells.	Hurtado-Bagès et al., 2020
mH2A1.2	Expression (mRNA) in C2C12 cells during growth and differentiation.	Dell’Orso et al., 2016 Posavec Marjanović et al., 2017
	Deposition is required for H3K27 acetylation and activation of muscle enhancers. Required for myogenic gene expression and myoblast differentiation.	Dell’Orso et al., 2016
	Inhibition of C2C12 myoblast fusion.	Hurtado-Bagès et al., 2020
H1b	Inhibition of C2C12 myoblast differentiation. Binding to Myod1 locus and inhibition of Myod1 transcription. Direct interaction with MSX1 at Myod1 locus.	Lee et al., 2004
	H1 variants are associated with the histone mark H3K9me3 in myogenic gene loci. Inhibition of C2C12 myoblast differentiation.	Chen et al., 2017

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
