# Peer review of "Epigenetic Regulation of Myogenesis: Focus on the Histone Variants"

_ijms, 2021, doi:10.3390/ijms222312727_

Round 1
Reviewer 1 Report
In this work Esteves de Lima and Relaix review and discuss the available literature on the role of histone variants in skeletal muscle myogenesis. The topic is certainly of high interest to the scientific community and in particular to those working in the muscle field. However, a recent excellent review by Nandini Karthik & Reshma Taneja (2020): Histone variants in skeletal myogenesis, Epigenetics, DOI: 10.1080/15592294.2020.1795606, has already covered this topic in a highly comprehensive way.
Given the similarity between these two reviews, I suggest adding new sections, for example on the role of histone variants in skeletal muscle disease, and/or novel technologies for the study of chromatin modifications.
The readability of this review is rather poor and would benefit from editing for English language and especially style. As it is, it reads like a long and dull list of information. The review could also be better organised by adding subsections (for example, I would have a section on Histone variants and chaperones (general intro) before proceeding to discuss their role in myogenesis.
Reviewer 2 Report
Perhaps the manuscript would benefit from some editing aimed at a better organisation of the information. In its present form, it is distributed into categories defined by the identity of the histone class: H3, H2A and linker H1.
Introducing Table I early to indicate that it summarises work related to the distinct variants that are going to be deployed, in detail, in the text that follows. Then, maybe the authors, once stated the large body of correlative evidence, showing, among other observations, the association with H3.3 with myogenic differentiation, may want to identify one or more mechanism impacted by turnover of histone variant together with the data supporting one or another. It may be of interest, too, to contemplate whether the impact of regulated histone variant deposition in myogenesis bear any specificity - even singular relevance - or it is similar to that in other cell types.
It is surprising that information regarding variants of different histone classes are treated as if they were totally independent. I don't know whether that reflects lack of information published or that it has not been integrated in a more comprehensive approach. Thus, Figs 1, 2 and 3 use regulatory regions of myogenic genes to illustrate functions of H3.3 and H2A.Z and H1 at similar sites, respectively. One could infer that the regulatory processes involved may be interrelated, but if not, it would merit to state it as such.
Aside the general comment, here are a few, minor examples of sentences illustrating the convenience of some editing.
Avoid repetitions
lines 102, 177 edit to avoid repetition
"H3.3 correlates with H3K4me3 and H3K36me3"
lines 166, 265
"In addition, H3.3 was linked to myogenic and neurogenic cell differentiation"
Clarification
lines 158-159
"At the myoblast stage Myod1 expression is 158 regulated by H3.3 deposition in the CER and in the promoter by the HIRA-ASF1A complex".
probably refers to changes in proximal nucleosomes; usually these regulatory regions are meant to be nucleosome free
lines 243, 244
"In fact, nucleosomes containing the subvariant H3mm7, that differs from H3.3 by only 2 aa, have a more unstable composition as observed by crystal structure analysis and by NaCl dissociation"
composition? or stability
lines 426-427
"Therefore, independently of age, obesity remains a large contributor to DNA damage in muscle, and combined with age it can have a more deleterious effect in the loss of muscle mass".
Although relevant to the paper quoted there, is hard to see how pertinent the statement is in the context of the review.
Reviewer 3 Report
The review "Epigenetic regulation of myogenesis: focus on the histone variants" written by Joana Esteves de Lima and Frédéric Relaix describes the role of histone variants in myogenesis and in particular in skeletal muscle development. Although more and more epigenetic mechanisms are revealed to be necessary in many differentiation processes, these mechanisms are rarely discussed in muscle development and even less those associated with histone variants or in related diseases. For this reason, I believe that this review has an important contribution to make in these areas of epigenetics and myogenesis. In my opinion, this review is very comprehensive and summarizes well the current knowledge in these fields.
I recommend that this review be published in the International Journal of Molecular Sciences (MDPI) with minor revision.
General Comments
In some cases, only recent work was referenced at the expense of older work. I suggest that original/older articles/reviews should also be referenced when older statements are involved.
For example:
Line 304-305: The H2A family is composed of several replication-independent variants in addition to the canonical H2A, such as H2A.Z, H2A.X, H2A.Bbd and macroH2A (mH2A) (Talbert and Henikoff, 2021).
I could suggest: Ausió, J. et al 2001 (https://doi.org/10.1139/o01-147)
Line 339 - 381: "Three distinct mH2A histones have been identified, mH2A1.1 and mH2A1.2, which are two isoforms generated by alternative splicing of the H2afy gene, and mH2A2 which is encoded by H2afy2 (Buschbeck et al., 2009)."
Some of the studies below seem more appropriate to me:
> Pehrson and Fried 1992 (DOI: 10.1126/science.1529340)
> Pehrson et al 1997 (DOI: 10.1002/(sici)1097-4644(199704)65:1<107::aid-jcb11>3.0.co;2-h)
> Chadwick and Willard, 2001 (DOI: 10.1093/hmg/10.10.1101) / Costanzi and Pehrson, 2001 (DOI: 10.1074/jbc.M010919200)
Body text
Line 41: "Epigenetic regulation of gene expression depends on chromatin architecture: post-translational modifications (PTMs) of histones, nucleosome composition and DNA methylation." Maybe the colon (:) after chromatin architecture meant to be a comma?
Line 400: It says "While mice lacking mH2A1 are viable and fertile without skeletal muscle defects". To my knowledge, although no major deficiency have been observed, no studies to date have specifically addressed skeletal muscle defects in mice lacking macroH2A. In contrast, several developmental and metabolic alterations have been observed (Pehrson et al., 2014).
Table
When indicating expression level, it would be very helpful to have the distinction between transcriptional and protein levels that may differ between species / during myogenesis and studies (e.g. H3.1 vs. H3.3 or mH2A1.2 vs. mH2A1.1)
There is a minor error where the second mH2A1.1 should be replaced by mH2A1.2.
For the mH2A isoforms, I suggest to add the results of the Posavec Marjanovic study ( DOI: https://doi.org/10.1038/nsmb.3481) in the table (the one related to their expression levels for example)
Figures
In general, I would suggest adding the reference of the studies to which the schemas are linked (either directly on the schema or with the legend).
Figure 2A
When indicating “H2A.Z mutated”, maybe it would be helpful to indicate what kind of mutation (maybe you mean the Ac-mut?)
Figure 2B
It would have been interesting to have a schema showing the opposing roles of mH2A1.2 and mH2A1.1 in fusion in parallel with the other schema presented.
I wish the authors good luck and I will be glad to see it published.
Best wishes.
Reviewer 4 Report
In this review, the authors summarized the latest findings on the role of histone variants (namely H3.3, H2A and H1 variants) in myogenesis. The title and abstract demonstrate the take-home message that readers should expect to get from this concise yet comprehensive review. The three schematics are simple yet informative, and the table adds to the value of the review.
I don't have any major comments except some minor corrections:
line 9: evidences -> evidence
ine 10: therefore, -> , therefore,
line 93: please refer interested readers to previous reviews that have discussed these points
Round 2
Reviewer 1 Report
The Authors have addressed all of my comments.